# Study of the Influence of Cutting Edge on Micro Cutting of Hardened Steel Using FE and SPH Modeling

**DOI:** 10.3390/mi13071079

**Published:** 2022-07-07

**Authors:** Lobna Chaabani, Romain Piquard, Radouane Abnay, Michaël Fontaine, Alexandre Gilbin, Philippe Picart, Sébastien Thibaud, Alain D’Acunto, Daniel Dudzinski

**Affiliations:** 1Applied Mechanics Department, FEMTO-ST Institute (UMR CNRS 6174), Bourgogne Franche-Comté University (UFC/ENSMM), 25000 Besançon, France; abnay.r@gmail.com (R.A.); michael.fontaine@femto-st.fr (M.F.); alexandre.gilbin@femto-st.fr (A.G.); philippe.picart@femto-st.fr (P.P.); sebastien.thibaud@femto-st.fr (S.T.); 2LEM3 Laboratory (UMR CNRS 7239), T-PRIOM Department, Lorraine University, Arts & Métiers ParisTech, 57070 Metz, France; romain.piquard@univ-lorraine.fr (R.P.); alain.dacunto@metz.ensam.eu (A.D.); daniel.dudzinski@univ-lorraine.fr (D.D.)

**Keywords:** micromachining, 41NiCrMo7 steel, finite element modeling, CEL, ALE, SPH modeling, cutting edge radius, cutting force, chip morphology

## Abstract

Micromachining allows the production of micro-components with complex geometries in various materials. However, it presents several scientific issues due to scale reduction compared to conventional machining. These issues are called size effects. At this level, micromachining experiments raise technical difficulties and significant costs. In this context, numerical modeling is widely used in order to study these different size effects. This article presents four different numerical models of micro-cutting of hardened steel, a Smooth Particle Hydrodynamics (SPH) model and three finite element (FE) models using three different formulations: Lagrangian, Arbitrary Eulerian–Lagrangian (ALE) and Coupled Eulerian–Lagrangian (CEL). The objective is to study the effect of tool edge radius on the micro-cutting process through the evolution of cutting forces, chip morphology and stress distribution in different areas and to compare the relevance of the different models. First, results obtained from two models using FE (Lagrangian) and SPH method were compared with experimental data obtained in previous work. It shows that the different numerical methods are relevant for studying geometrical size effects because cutting force and stress distribution correlate with experimental data. However, they present limits due to the calculation approaches. For a second time, this paper presents a comparison between the four different numerical models cited previously in order to choose which method of modeling can present the micro-cutting process.

## 1. Introduction

Micro-technologies are taking over the whole world. They have become essential in all fields like medical, luxury, tools production, aeronautics and watchmaking. Micromachining is one of the processes that allow the obtaining of micro components with complex geometries in variable materials with low tolerance. Otherwise, cutting at the micro-scale differs from the conventional one. It presents many scientific and technological issues due to scale reduction.

Among these issues, called size effect there is:The influence of cutting edge radius

In reduced scale, the tool edge has a rounded shape less sharp than in conventional cutting, which affects the integrity of the finished surface [1,2].
Work-piece material microstructure


The microstructure of the work material can also affect micro-cutting. During micromachining, the tool edge radius is in the same range as the material grain size, for example, “a smaller material grain size resulted in a larger cutting force and a higher specific cutting energy” [3].
Multistep cutting


According to Xuhao Song et al. [4], multistep cutting increases the chip segmentation degree of the second step, it also increases the average cutting force of the second step slightly. This is due to accumulated residual stress on the previous machined surface. On the other hand, during micro-cutting, especially in micro-milling and broaching, exit burrs can be accumulated after each cut. Many experimental studies were conducted to understand this phenomenon [5,6] but a numerical model is not used yet to understand accumulation in burr formation during micro-cutting.

Knowing that experimental studies are expensive and take a lot of time, researchers have used numerical simulation in order to understand the material removal in micro-cutting. Most numerical models have been developed within finite difference method, finite element method and meshless methods [7]. These methods were used by several works to study orthogonal micro-cutting [8]. This work presents a comparison between different numerical models of orthogonal micro-cutting of hardened steel 41NiCrMo7. The purpose is to determine which model can better simulate the micro-cutting process in order to study multi-pass micro-cutting, exit burr formation and tool geometry effect.

The paper is divided into two parts. The first part presents a comparison between two different numerical models: meshless method, Smooth Particle Hydrodynamics (SPH) and finite element (FE) model using the Lagrangian formulation in order to analyze their capacity to predict the size effect due to the ratio between tool edge radius and depth of cut. The second part of the work, presents a comparison between four numerical models: meshless method, SPH and three different FE models using three different formulations: Lagrangian, Arbitrary Eulerian–Lagrangian (ALE) and Coupled Eulerian–Lagrangian (CEL). In all models, the evolution of cutting forces, chip formation and stress distribution are predicted and compared to experimental results. Results show that, firstly, FE modeling can be a great tool to understand different phenomena in metal micro-cutting. Secondly, CEL and ALE formulation better describe better micro-cutting since they combine the Eulerian and Lagrangian approaches.

## 2. Numerical Modeling of Orthogonal Micro-Cutting

### 2.1. SPH Model

A 3D SPH model is developed to study the orthogonal cutting mechanic. This type of modeling uses a particular method: The work-piece is presented by a mesh of particles. Each particle will have an influence in a neighborhood. The influence depends on the distance between the particles described by a weighting function *W*. The advantage is that no remeshing is necessary and the damage occurs naturally when the particles leave the zones of influence. The main drawback is the management of free borders, where the particles have a neighborhood. As a result, state variables are not optimally evaluated.

To simulate the cutting process with different tool geometry properly, it is necessary to introduce a material flow stress model to describe the material behavior. The traditional Johnson–Cook material constitutive law with material damage criterion was used for the proposed model. The Johnson–Cook model is described by the expression of average flow stress given by Equation (1).
(1)$$\sigma =\left[A+B\;\varepsilon^{n}\right]\times \left[1+C\ln\frac{\dot{\varepsilon }}{\dot{\varepsilon_{0}}}\right]\times \left[\;1-\left(\frac{T-T_{room}}{T_{melt}-T_{room}}\right)\right]$$
where ε is the equivalent plastic strain, $\dot{\varepsilon }\;$and $\dot{\varepsilon }$_0_ are the equivalent and reference plastic strain rates, *T*, *T_melt_* and *T_room_* are the local, melting and room temperatures, respectively, *n* is the strain hardening index and m is the thermal softening index. Johnson–Cook Parameters *A*, *B* and *C* represent the yield strength, strain and strain rate sensitivity of the material. For reasons of data availability, simulations are carried out with a 41NiCrMo7 steel, hardened to 41 HRC.

Table 1 shows physical and mechanical properties of the steel hardened to 41 HRC [9].

Table 2 shows the Johnson–Cook’s law parameters related to steel hardened at 41 HRC [10].

A method of penalization is used to take into account the contact. When a slave element (particle of the work-piece) penetrates a master element by a certain distance along the normal to the contact, a contact force equivalent to a spring working only in tension is defined and tends to eject the penetrating element. Friction is represented by a Coulomb model [11]. The work-piece is subjected to boundary conditions in displacement on the lower free edge (fixed support) and on the left edge (zero displacement in X direction). The sides are also constrained in displacement (zero displacement in Z direction). The length *L* is 200 µm, the cutting width *w* is 5 µm and the height defined under the tool is 20 µm. The particle density is homogeneous and is characterized by a square mesh arrangement with a distance between particles of *d* = 1 µm, see Figure 1.

Damage criterion is not necessary in SPH modeling as the particles move apart during chip formation, they move away from the zones of influence during separation. The damage is then intrinsic to the SPH resolution. On the other hand, SPH modeling requires an equation of state in a polynomial form (Equation (2)), which defines the pressure *P* as a function of the variation in volume *μ* of the system.
(2)$$P=C_{0}+C_{1}\mu +C_{2}\mu^{2}+C_{3}\mu^{3}+\left(C_{4}+C_{5}\mu +C_{6}\mu^{2}\right)E$$

Only the parameter *C_1_* is non-zero and corresponds to the incompressibility modulus *K* of the material, considered isotropic. This modulus is related to the Young modulus *E* and to the Poisson’s ratio *ν* by the following Equation (3).
(3)$$K=\frac{E}{3\left(1-2\nu \right)}$$

### 2.2. FE Model Using Lagrangian Formulation

A 2D orthogonal micro-cutting FE model based on the Lagrangian formulation is developed using ABAQUS software with an explicit dynamic integration scheme to analyze the cutting mechanism. The use of a Lagrangian formulation requires the use of a separation criterion between the machined part and the chip in finite element modeling. The cutting tool used for the modeling purpose is considered a rigid body and located by a reference point (RP) to acquire the cutting force value.

For realistic modeling, a cutting angle of *ɣ* = 8°, a clearance angle of α = 6° and a cutting edge radius of 1.5; 2; 5; 8; 10 (µm) are applied to the cutting tool. The cutting velocity *V_c_* applied to the cutting tool corresponds to the experimental setup (*V_c_* = 40 m/min) and a depth of cut *h_c_* = 4 µm is used in the simulations. The tool/part interaction is modeled with a surface contact (“surface to surface contact” in ABAQUS) with Coulomb friction law as defined in the bibliography [12] and the coefficient of friction attributed in this case is *f* = 0.63 [11].

A fine mesh is applied in the zone of strong deformations with quadratic first-order elements with a reduced integration, see Figure 2. The use of this type of element can lead to zero energy strain modes, called “Hourglass” modes, which must be eliminated. It is commonly admitted in the literature that the artificial energy imposed on the elements to prevent Hourglass modes should not exceed 10% of the internal energy of the model [13].

In this context, a comparative study between the different management methods of “Hourglass” available in the software led to the choice of the “Combined” one because it imposes the most significant artificial energy on the elements and it gives a greater radius of curvature of the chip compared to the other modes.

Work-piece material flow stress is described by the traditional Johnson–Cook material constitutive law where the equation and parameters are presented in Section 2.1.

In order to simulate the chip formation, a failure criterion is used. It is described by the Johnson–Cook failure model. This model takes into account the influence of strain, strain rate and temperature on material failure shown in Equation (4). For simplification reasons, temperature influence is neglected. The failure parameters of the Johnson–Cook model [10] are shown in Table 3.
(4)$$\varepsilon =\left[D_{1}+D_{2}\;\exp\left(-D_{3\;}\eta \right)\right]\times \left[D_{4}+\ln\frac{\dot{\varepsilon }}{\dot{\varepsilon_{0}}}\right]\times \left[1+D5\left(\frac{T-T_{room}}{T_{melt}-T_{room}}\right)\right]$$

### 2.3. FE Modeling Using Combined Formulations: ALE and CEL

Many approaches are used in literature in order to develop a FE model of the metal orthogonal cutting process. Initially Eulerian and the Lagrangian methods were widely used.

In the Eulerian approach, the mesh is fixed and the material moves through it. The utilization of this method requires a predefined chip shape to develop the simulation, but a correct assumption of the chip shape is difficult to obtain.

In the Lagrangian approach, the mesh is linked to the material so it deforms with the material. This leads to severe element distortion. Otherwise, in order to simulate chip formation during cutting a chip separation criterion needs to be applied to the model. It requires also the activation of the element deletion algorithm, in which elements with satisfying maximal plastic strain or damage are deleted. In this case, the study of superficial residual stresses will be difficult.

In order to avoid the disadvantages of the Eulerian and the Lagrangian methods FE models of orthogonal cutting combining these two approaches are developed. In this paper, two FE models—Arbitrary Lagrangian–Eulerian approach (ALE) and Coupled Eulerian–Lagrangian approach (CEL)—were developed.

In both models, the work-piece is made of hardened steel. Its mechanical and physical properties are mentioned in Section 2.1. The tool, represented by a rigid body, is made of tungsten carbide and its behavior is described by a linear elastic law. Its physical and mechanical properties are mentioned in Table 4 [14]. To describe material flow during simulation, the Johnson—Cook law is used. The constitutive equation and its parameters are already mentioned in Section 2.1. To be in coherence with the previous models, the temperature effect was neglected in this model (*T* = *T_room_* in Equation (1)).

#### 2.3.1. ALE Model

A FE model of orthogonal cutting using the Arbitrary Eulerian–Lagrangian approach (ALE) is developed. In this approach, Eulerian and Lagrangian boundaries were used [15], see Figure 3.

The tool was initially motionless in all directions and then animated by the cutting velocity (see Figure 3). The cutting velocity *V_c_* applied to the cutting tool corresponds to the experimental setup (*V_c_* = 40 m/min) and a depth of cut *h_c_* = 4 µm is used in the simulations. A cutting angle of *ɣ* = 8°, a clearance angle of α = 6° and a cutting edge radius of 2 µm are applied to the cutting tool.

To simulate interactions between the tool and the work-piece, the Coulomb friction law is adopted with a friction coefficient *f* = 0.63 [11]

Figure 4 shows the chip shape and the distribution of Von Mises stress. We can see the different deformation zones. We notice that elements are elongated. At the end of the chip elements, the movement is not enough, which does not help with mesh distortion. We notice that calculations end immediately because of high mesh distortion. In the Eulerian approach, a chip shape needs to be predefined which is a major disadvantage, but on the other hand, it allows for a longer time of cutting process modeling.

To overcome these disadvantages, Ducobu et al. [16] used the coupled Eulerian–Lagrangian approach (CEL) in the modeling of orthogonal cutting of titanium alloy and compared the results to those of ALE and experimental results. They concluded that using the CEL approach in metal cutting modeling guaranteed the accuracy of results and improved simulation parameters. From our side, a 3D FE model of orthogonal micro-cutting of hardened steel with a carbide tool using the CEL approach is developed under Abaqus/Explicit software in order to check its capability to combine advantages inherent to each method by avoiding their drawbacks.

#### 2.3.2. CEL Model

CEL method was first used to model fluid-structure interactions. Then in 2011, it was suited for problems that involve large deformation, very sensitive to mesh [17]. It was used then to model metal forming problems [18], friction tests modeling and for calibration of used friction models [14]. We propose to use it to model orthogonal cutting.

In this type of modeling, an Eulerian mesh represents a volume in which the Eulerian material (work-piece) flows and interacts with the Lagrangian part(s) (Tool) (see Figure 5).

The Eulerian mesh should be large enough to cover the Eulerian material and the formed chip. As shown in Figure 6, EVF (Element Void Fraction) measures how the element is filled with material.

EVF = 1, the element is empty.EVF = 0, the element is full.

In order to simulate interactions between the tool and the work-piece, the Coulomb friction law is adopted with a friction coefficient *f* = 0.63 [11] and a penalty contact algorithm is activated. The work-piece is subjected to zero speed boundary conditions in order to fix it in space, see Figure 5. The tool was initially motionless in all directions and then animated by the cutting velocity (see Figure 5). The cutting velocity *V_c_* applied to the cutting tool corresponds to the experimental setup (*V_c_* = 40 m/min) and a depth of cut *h_c_* = 4 µm is used in the simulations. For realistic comparison, a cutting angle of *ɣ* = 8°, a clearance angle of α = 6° and a cutting edge radius of 2 µm are applied to the cutting tool.

To help calculation, a part of the Lagrangian zone (tool) that is supposed to interact with the Eulerian zone (work-piece) must be placed in the Eulerian mesh at the beginning of the simulation, see Figure 6. 3D 8-node rectangular elements were used to mesh the work-piece and the tool.

## 3. Results

### 3.1. Experimental Set-Up

Elementary micro-cutting machining tests were carried out on a Röders RP600 CNC, which performed adequately to ensure minimal errors for the cutting speed, feed rate and position (see Figure 7). The chip conveyor was switched off to minimize cutting force signal noise resulting from vibrations. As shown in Figure 3, three tools were used as follows: two to obtain the tube dimensions directly before beginning each test from a larger tube to minimize tube run-out and the last to perform the experiments. Force measurements were carried out using a Kistler Minidyn 9256C2 dynamometer associated with a Kislter amplifier 9017. Data were saved using a National CompactDaq 9174 with NI9215 cards and a dedicated Labview program [11].

Reference numerical simulations use the same cutting conditions and tool geometry as the performed elementary micro-cutting machining tests. However, The work-piece was made of 40NiCrMo16 steel hardened at 54HRC. The material used for the numerical simulations is AISI 4340 steel, different from the steel used in the elementary micro-cutting tests because no constitutive law is known for 45NiCrMo16 steel. It is close in composition (41NiCrMo7) and hardness (47 HRC) to 45NiCrMo16 (54 HRC), so it is assumed that the behavior of these two materials is similar.

### 3.2. Study of the Cutting Edge Radius Influence Using SPH and Lagrangian Models

In order to study the influence of the $h_{c}/r_{\beta }$ ratio on the cutting process, a cut angle *ɣ* = 8° is fixed. The cutting edge radius *r_β_* and the cutting height *h_c_* are set at 2–5–8–10 µm and 2–4 µm respectively, as shown in Table 5. Observation of the von Mises equivalent stress shows that whatever the cutting depth or the cutting edge radius, the maximum stress is in order of 1500 MPa in all cases.

On the other hand, the chip morphology and the stress field distribution are different depending on the case. Simulation results with a larger edge radius show that the shear zones merge to form a single zone. This explains the increase in the feed force *F_f_* for cases where the $h_{c}/r_{\beta }$ ratio is less than one see Figure 8. As a result, the material is hardened and the stress rises in front of the cutting edge, and in consequence, cutting forces rise as well.

For the SPH method, the main studied parameters were the edge radius, the cutting depth and the cutting angle. Only three values of *h_c_* (2, 11, 20 µm) and γ (−8°, 0° et 8°) were studied. The *h_c_* parameter varied between 1 and 14 µm. As noticed in the FE method, the maximum von Mises stress of 1320 MPa is reached in almost all cases. Figure 9 shows the evolution of cutting forces (in absolute value) in the different cases (various values of edge radius, rake angle and cutting depth). Several points should be mentioned. Firstly, *r_β_* has a large influence on the cutting forces. Secondly, when *h_c_* is smaller than *r_β_* (0.5 *h_c_* ≤ *r_β_* ≤ *h_c_*) force curves overlap, which means that the cutting angle has no influence on cutting forces for small cutting depths *h_c_*. For higher values of *h_c_*, cutting forces depend on cutting angle, i.e., cutting forces increase when cutting angle decreases. Finally, for important values of *r_β,_* the cutting force component is lower than the feed force component (mainly when the cutting angle is no longer influencing). We can notice the independence of cutting forces from rake angle by plotting the obtained results as a function of cutting depth and cutting edge radius in Figure 10. It is clear that for a ratio $h_{c}/r_{\beta }$ less than 0.8, the feed force *F_f_* is greater than the cutting force *F_c_*. Likewise, the cutting angle no longer influences forces for a ratio $h_{c}/r_{\beta }$ lesser than 1 and expresses a predominant plowing regime. Concerning the stress distribution observed from SPH simulations, we can notice that the dimensionless parameter$\;h_{c}/r_{\beta }$ provides results completely independent of the edge radius value. Figure 11 illustrates the geometry of the chip formation zone through the maximum of Von Mises stress as a function of$\;h_{c}/r_{\beta }$. We can see for low ratios of $h_{c}/r_{\beta }$, that the zone starts at the lowest point of the tool and then takes a comma shape without following the radiated part of the tool, suggesting the appearance of a dead zone. For the higher ratios, we obtain a band shape, which is common in the literature. It should be noticed that with the SPH model, the material flow near the cutting edge is highly reduced with important values of $r_{\beta }$. It corresponds to experimental observations made from quick stop tests in micro-cutting where a sticking phenomenon is clearly identified in this region of the chip for high values of edge radius [11].

The two models show as outputs the distribution of von Mises stress, cutting force and chip shape in order to study the influence of tool geometry, especially the cutting edge radius on th ecutting process. Results show that in micro-cutting, maximum cutting force and Von Mises stress values are proportionally greater than in conventional machining. This is due to the reduction in the dimensions of the tool in regard to cutting depth.

With the FE model using a Lagrangian formulation, the cutting forces correspond well to the experimental results. High sensitivity to the *h_c_*/*r_β_* ratio was observed from the stress field distribution and the chip morphology is consistent. We noticed that force values were higher than the experimental forces due to excessive element distortion using the Lagrangian approach.

The SPH method shows the same tendencies concerning cutting force and stress evolution but presents more discrepancies with the experimental data, mainly on cutting force values and chip morphology. Nevertheless, the material flow seems to be more realistic around the cutting edge than in the FE model [19].

We can say that the two methods show the capability to reproduce the *h_c_*/*r_β_* scale effect and allow studying the evolution of force and stress zones and even material flow, but not with the same relevance on each parameter. However, they show limitations that could be avoided by using a new FE modeling approach.

### 3.3. Models Comparison

Four numerical models (SPH, Lagrangian, ALE and CEL) simulate orthogonal micro cutting of hardened steel with a tungsten carbide tool under the same cutting conditions (The cutting velocity *V_c_* = 40 m/min, depth of cut *h_c_* = 4 µm, a cutting angle of *ɣ* = 8°, a clearance angle of α = 6° and cutting edge radius of 2 µm). The purpose is to compare this different methods in order to decide which of these methods is the most relevant to study size effects in the case of multi-pass micro-cutting.

All models illustrate stress distribution during micro-cutting. We can see the different deformation zones see Figure 12:
Primary deformation zone: Result of the high deformation of the material under high strain rates conducting to chip formation initiation.Secondary deformation zone: Due to the contact between the rake face and the chip.Tertiary deformation zone: Due to the contact between the flank face and the machined surface.


Figure 13 shows that the numerical chip is continuous whatever the type of simulation. The chip morphology obtained from numerical models is more realistic compared to the SPH model. We notice that the chip shape simulated by the CEL model is closer to the experimental chip shape [11] and its formation on the tip of the tool is better presented compared to the Lagrangian model.

The quality of mesh in the chip and the machined surface is different. In the Lagrangian model, we can observe elongated elements due to mesh deformation. For the ALE model, we notice that elements are elongated too. At the end of the chip elements, movement is not enough which does not help with mesh distortion. We notice that calculations end immediately because of high mesh distortion. In the Eulerian approach, a chip shape needs to be predefined which is a major disadvantage, but on the other hand, it allows for a longer time for cutting process modeling.

The problem of mesh elongation is avoidable in the CEL method thanks to the fact that the work-piece is presented by an Eulerian mesh representing a volume in which the Eulerian material flows and interacts with the Lagrangian part which is the tool.

Von Mises stress in the Lagrangian model is high compared to the other models due to excessive element distortion.

As for the chip shape, cutting forces and feed forces recorded during cutting, numerical simulations are closer to the experimental values, which are slightly higher due to the vibration of the machining system, tool holder and dynamometer, not taken into account in the model. See Table 6.

In Table 6, we can notice that cutting and feed forces for different finite element models are close to experimental values, which means the finite element can represent micro-cutting at the micro-scale. SPH model presents more discrepancies with the experimental data, mainly on cutting force values.

## 4. Discussion

After testing these different methods to simulate the orthogonal micro-cutting process of stainless steel, it is possible to comment on their relevance in order to study the different aspects of micro machining, especially in the context of multi-pass micro-cutting. The models are here compared according to the following outputs: distribution of von Mises stress, cutting forces and chip shape.

The influence of cutting edge radius on the cutting process is studied using two different numerical models (Lagrangian and SPH). Results show that in micro-cutting, maximum cutting force and von Mises stress values are proportionally greater than in conventional machining. This is due to the reduction in the dimensions of the tool in regard to cutting depth. With the FE model using a Lagrangian formulation, the cutting forces correspond well to the experimental results. High sensitivity to the $h_{c}/r_{\beta }$ ratio was observed from the stress field distribution and the chip morphology is consistent. The SPH method shows the same tendencies concerning cutting force and stress evolution but presents more discrepancies with the experimental data, mainly on cutting force values and chip morphology. Nevertheless, the material flow seems to be more realistic around the cutting edge than in the FE model. The two methods show the capability to reproduce the $h_{c}/r_{\beta }$ scale effect and allow studying the evolution of force and stress zones and even material flow, but not with the same relevance on each parameter.

Chip shape and cutting forces in CEL and ALE models are closer to the experimental results compared to the other methods. Mesh distortion was absent with the CEL method contrary to the ALE model where distortions led to premature termination of computations. Finally, the most important point during CEL simulation is that the element deletion algorithm is deactivated, which makes possible the extraction of information such as residual stresses on the machined surface.

## 5. Conclusions and Prospects

This paper presents different methods to simulate the orthogonal micro-cutting process. The objective is to find the method that provides the most relevant results in order to study the different aspects of micro machining, especially in the context of multi-pass micro-cutting. In this view, four different numerical models simulating metal cutting during orthogonal micro-cutting of hardened steel 41 HRC were presented in this work.

Results show that numerical modeling is relevant to studying micro-cutting mechanisms. It provides information about chip shape, stress distribution, cutting forces, plastic strain and chip shape. Comparison between different formulations led us to consider that CEL formulation could be a promising method to model micro-cutting and especially multistep micro-cutting thanks to its numerous advantages:
Better quality of machined surface since element deletion algorithm is deactivated.No longer distortion problems.Numerical chip shape is close the experimental one.Different deformation zones are identifiable.


Further studies need to be carried out in order to optimize CEL models for the purpose of simulating properly the micro-cutting process by taking into account successive passes of the cutting edge into the machined material. In fact, numerous materials in the field of micro-machining present a high hardening capacity and because of the small depths of cut used in micro machining, studying size effects and resultant surface integrity cannot be conducted without considering multi-pass cutting. It is a fact that most metals submitted to micro-cutting process tend to accumulate plastic stress on the machined surface and that can lead to change in chip formation and an increase in cutting forces for small depths of cut. It can cause burrs accumulation at tool exit zones too.

For this purpose, it is necessary to identify a new friction model to better describe the frictional behavior on tool-work-piece-chip interfaces by combining experimental friction tests and numerical friction models. To obtain more accurate results, the effect of the variation of temperature should be taken into account in the constitutive and friction laws. Moreover, micro-cutting quick-stop tests could be conducted in order to compare the chip shape and deformation zones with numerical data.

## Figures and Tables

**Figure 1 micromachines-13-01079-f001:**
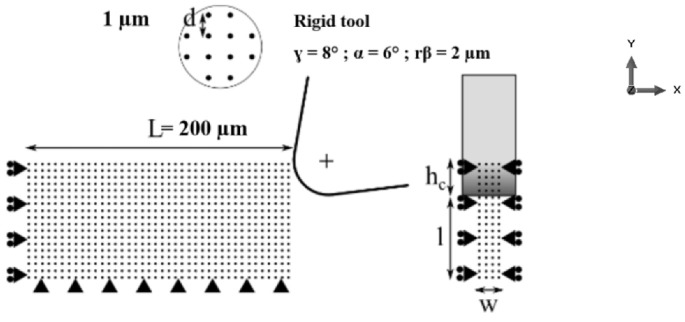
SPH model of orthogonal micro cutting: geometry; boundary conditions and mesh.

**Figure 2 micromachines-13-01079-f002:**
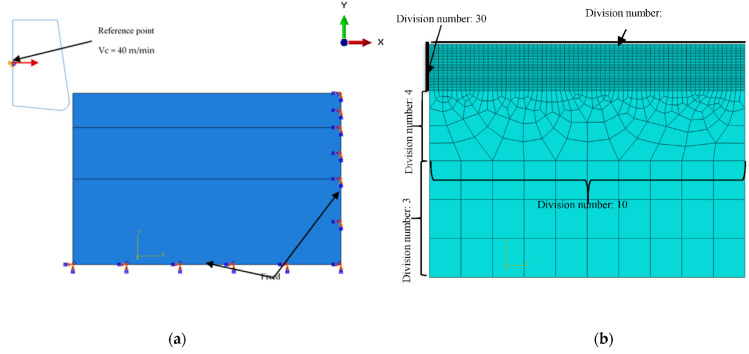
Finite element model of orthogonal micro-cutting: (**a**) Geometry and boundary conditions; (**b**) Mesh.

**Figure 3 micromachines-13-01079-f003:**
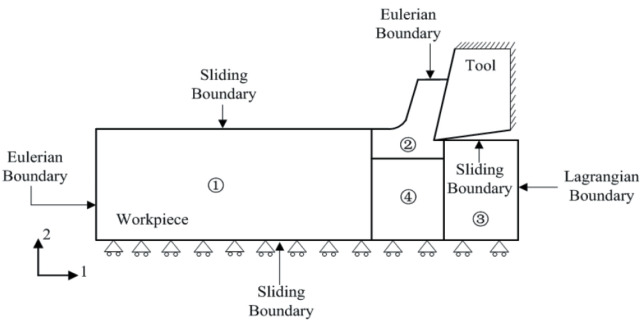
Description of the ALE model.

**Figure 4 micromachines-13-01079-f004:**
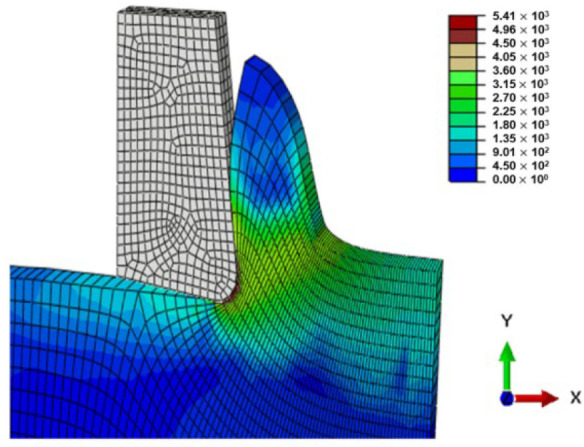
Von Mises stress distribution during micro-cutting (ALE model).

**Figure 5 micromachines-13-01079-f005:**
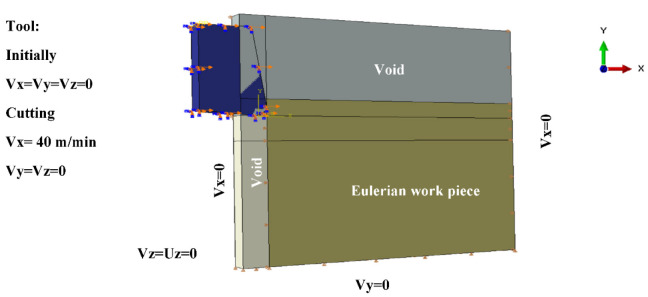
CEL model description: Geometry and boundary conditions.

**Figure 6 micromachines-13-01079-f006:**
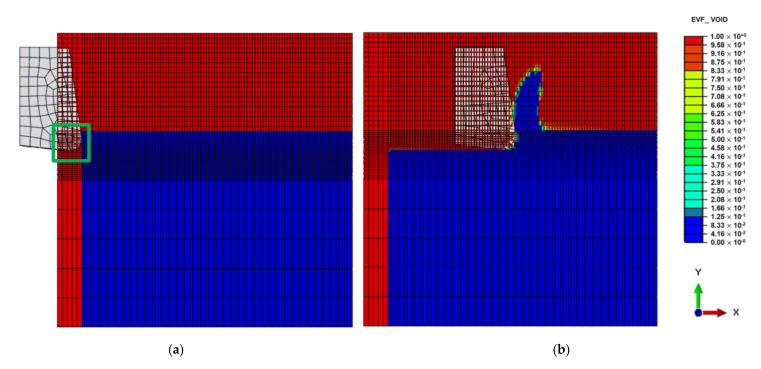
Element void fraction: (**a**) Initially; (**b**) During cutting.

**Figure 7 micromachines-13-01079-f007:**
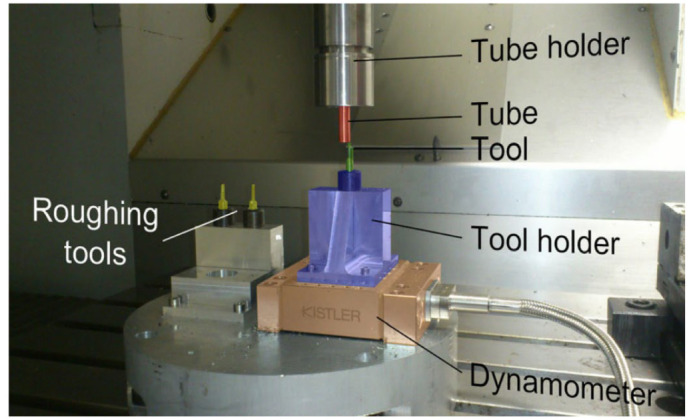
Experimental set-up for tube turning.

**Figure 8 micromachines-13-01079-f008:**
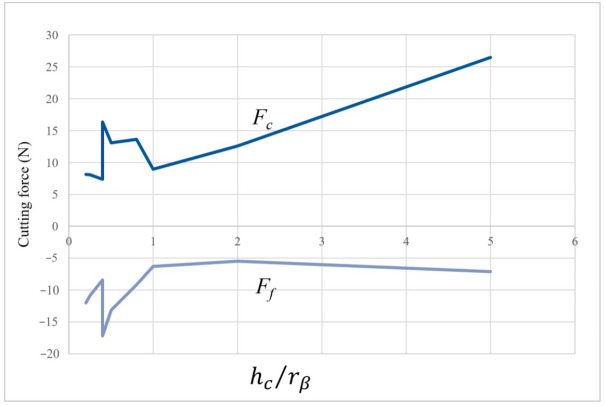
Cutting force evolution with $\frac{h_{c}}{r_{\beta }}$ ration.

**Figure 9 micromachines-13-01079-f009:**
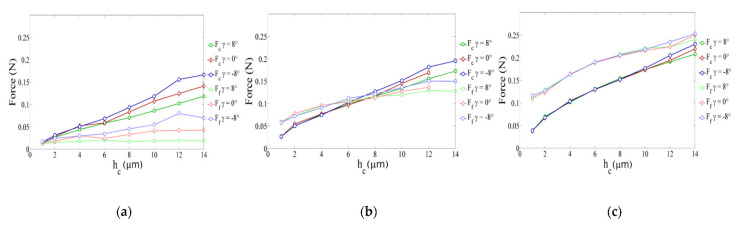
Cutting force evolution for different cutting depth *h_c_* and cutting edges: (**a**) for *r_β_* = 2 µm; (**b**) for *r_β_* = 11 µm; (**c**) for *r_β_* = 20 µm.

**Figure 10 micromachines-13-01079-f010:**
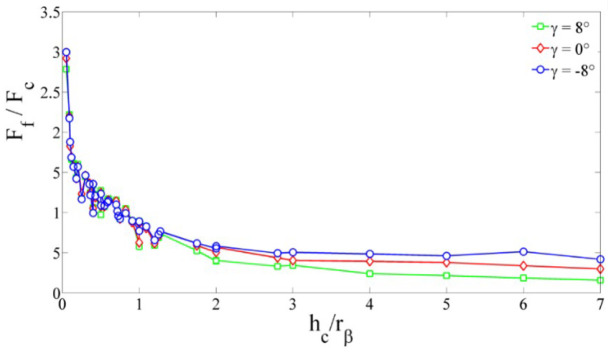
Evolution of $F_{f}/F_{c}\;$with $h_{c}/r_{\beta }$.

**Figure 11 micromachines-13-01079-f011:**
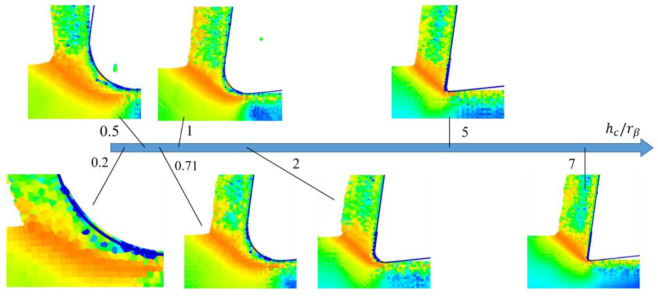
Evolution of the geometry of the chip formation zone through the maximum of Von Mises stress as a function of $h_{c}/r_{\beta }$.

**Figure 12 micromachines-13-01079-f012:**
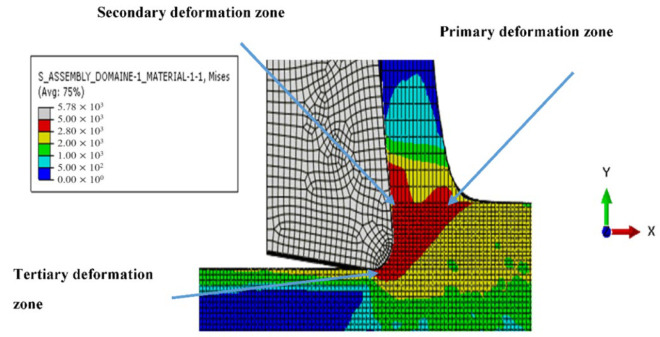
Different deformation zone during micro orthogonal cutting.

**Figure 13 micromachines-13-01079-f013:**
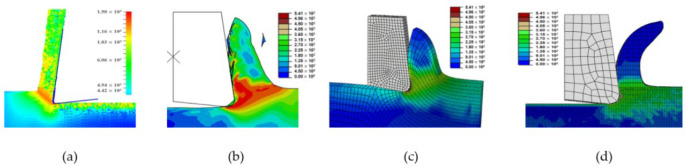
Chip shape and Von Mises stress distribution during orthogonal micro-cutting: (**a**) SPH; (**b**) Lagrangian; (**c**) ALE; (**d**) CEL.

**Table 1 micromachines-13-01079-t001:** Physical and mechanical properties of 41NiCrMo7.

Material	Density	Young Modulus	Poisson Coefficient
41NiCrMo7	7.85 g/cm^3^	207 GPa	0.2

**Table 2 micromachines-13-01079-t002:** Johnson cook’s law parameters related to 41NiCrMo7.

Material	A	B	C	*n*
41NiCrMo7	792 MPa	510 MPa	0.014	1.02

**Table 3 micromachines-13-01079-t003:** Failure parameters of the Johnson–Cook model for 41NiCrMo7.

*D* _1_	*D* _2_	*D* _3_	*D* _4_
0.05	3.44	−2.12	0.002

**Table 4 micromachines-13-01079-t004:** Mechanical and physical properties of WC-Co.

Material	Density	Young Modulus	Poisson Coefficient
WC-Co	1.48E−8 t/mm^3^	368 GPa	0.25

**Table 5 micromachines-13-01079-t005:** The influence of *h_c_* and *r_β_* on chip formation.

	*h_c_* = 2 µm	*h_c_* = 4 µm
*r_β_* = 2 µm	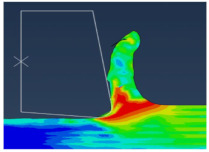	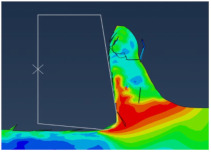
*r_β_* = 5 µm	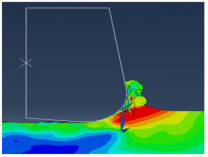	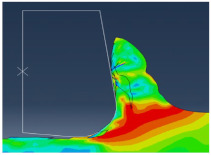
*r_β_* = 8 µm	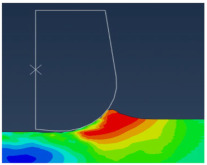	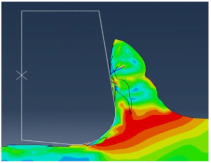
*r_β_* = 10 µm	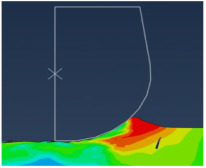	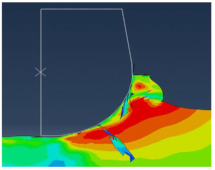

**Table 6 micromachines-13-01079-t006:** Cutting and feed forces for different methods.

	Experimental	SPH	Lagrangian	ALE	CEL
*F_c_* (N)	8	3,2	12	6	7
*F_f_* (N)	−7	−3	−5	−4	−5

## Data Availability

Not applicable.

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
