# Peer review of "Study of the Influence of Cutting Edge on Micro Cutting of Hardened Steel Using FE and SPH Modeling"

_micromachines, 2022, doi:10.3390/mi13071079_

Round 1

Reviewer 1 Report

The article has issues for which I request authors to look at following points: 1. The introduction is not clear and does not provide exact information that what is trying to be said. 2. After looking at the 12 references only, I observed that these are very less for a good paper and none of the journal belongs to 2020 or 2021 and so on... due to this, I doubt author has gone through latest work in the field which gives possibility of better work than their own work in literature. 3. The explanations consider many assumptions or hypothesis which means there is no guarantee that work that authors have presented is actually correct. In the light of above points, I do not feel that this article is novel and can add significant knowledge to particular field.

Reviewer 2 Report

See annex for details

Reviewer 3 Report

The manuscript entitled "Study of the influence of cutting edge on micro cutting of hardened steel using Fe and Sph modelling" discussed the influence of cutting edge in micro-cutting under size effect. The topic is interesting. The comments are presented as below:

  1. For the title, the "Fe" and "Sph" should be replaced by the "FE" and "SPH". What ARE the "Fe" and "Sph"?
  2. The paper seems confused, which should be re-written.
  3. In the ABSTRACT, WHAT are the four models?
  4. In the INTRODUCTION, more papers should be cited.
  5. The Section 2 should not only include finite element and SPH models but also ALE and CEL methods.
  6. In the RESULTS, the Section 3.1 should be a new section "Experiment Setup".
  7. Check Figure 13 and Figure 14, and more information has been lost.
  8. The paper should include the section "Discussion".
  9. The "Conclusions and prospects" is too long.
  10. What is the contribution and significance?

Round 2

Reviewer 1 Report

The revisions are not acceptable as the responses are not sufficient and look more like excuses.

In my opinion, article is not as per standard of this journal of repute.